# AMPK-Dependent YAP Inhibition Mediates the Protective Effect of Metformin against Obesity-Associated Endothelial Dysfunction and Inflammation

**DOI:** 10.3390/antiox12091681

**Published:** 2023-08-28

**Authors:** Lijing Kang, Juanjuan Yi, Chi-Wai Lau, Lei He, Qinghua Chen, Suowen Xu, Jun Li, Yin Xia, Yuanting Zhang, Yu Huang, Li Wang

**Affiliations:** 1Department of Biomedical Sciences, City University of Hong Kong, Hong Kong 999077, China; lijikang@cityu.edu.hk (L.K.); leihe8@cityu.edu.hk (L.H.); qinghchen2-c@my.cityu.edu.hk (Q.C.);; 2School of Biomedical Sciences, The Chinese University of Hong Kong, Hong Kong 999077, China; chiwailau@cuhk.edu.hk (C.-W.L.); xia.yin@cuhk.edu.hk (Y.X.); 3Hong Kong Center for Cerebro-Cardiovascular Health Engineering (COCHE), Hong Kong 999077, China; 4Department of Infectious Diseases and Public Health, City University of Hong Kong, Hong Kong 999077, China; juanjuyi-c@my.cityu.edu.hk (J.Y.); jun.li@cityu.edu.hk (J.L.); 5Department of Endocrinology, The First Affiliated Hospital of USTC, Division of Life Sciences and Medicine, University of Science and Technology of China, Anhui 230027, China; sxu1984@ustc.edu.cn; 6Department of Electronic Engineering, The Chinese University of Hong Kong, Hong Kong 999077, China; ytzhang@cuhk.edu.hk

**Keywords:** high glucose, AMPK, YAP, JNK, endothelial dysfunction, obesity, metformin

## Abstract

Hyperglycemia is a crucial risk factor for cardiovascular diseases. Chronic inflammation is a central characteristic of obesity, leading to many of its complications. Recent studies have shown that high glucose activates Yes-associated protein 1 (YAP) by suppressing AMPK activity in breast cancer cells. Metformin is a commonly prescribed anti-diabetic drug best known for its AMPK-activating effect. However, the role of YAP in the vasoprotective effect of metformin in diabetic endothelial cell dysfunction is still unknown. The present study aimed to investigate whether YAP activation plays a role in obesity-associated endothelial dysfunction and inflammation and examine whether the vasoprotective effect of metformin is related to YAP inhibition. Reanalysis of the clinical sequencing data revealed YAP signaling, and the YAP target genes CTGF and CYR61 were upregulated in aortic endothelial cells and retinal fibrovascular membranes from diabetic patients. YAP overexpression impaired endothelium-dependent relaxations (EDRs) in isolated mouse aortas and increased the expression of YAP target genes and inflammatory markers in human umbilical vein endothelial cells (HUVECs). High glucose-activated YAP in HUVECs and aortas was accompanied by increased production of oxygen-reactive species. AMPK inhibition was found to induce YAP activation, resulting in increased JNK activity. Metformin activated AMPK and promoted YAP phosphorylation, ultimately improving EDRs and suppressing the JNK activity. Targeting the AMPK–YAP–JNK axis could become a therapeutic strategy for alleviating vascular dysfunction in obesity and diabetes.

## 1. Introduction

Obesity is an independent risk factor for cardiovascular disease [1]. Increased blood glucose (hyperglycemia) is associated with endothelial dysfunction in obesity [2,3]. Endothelial inflammation is closely associated with obesity and diabetes [2,4]. The overexpression of pro-inflammatory cytokines in obesity is considered to be a link between obesity-associated inflammation and endothelial dysfunction [3]. Hyperglycemia, insulin resistance, oxidative stress, etc., play critical roles in endothelial dysfunction in obesity [5]. However, the exact mechanisms have not been fully elucidated. Recent studies have revealed that the diminished activity of AMPK plays a significant role in hyperglycemia-associated endothelial dysfunction [6,7], although the detailed mechanisms are not fully understood.

YAP is the effector of the Hippo pathway, a highly conserved kinase cascade that controls organ size, cell contact, cancer metastasis, and tissue homeostasis [8,9]. Previous studies showed that YAP activation in the vascular endothelium plays a critical role in disturbed flow-induced atherogenesis through JNK activation [10,11]. We hypothesize that YAP activation participates in endothelial dysfunction. YAP was reportedly activated in multiple organs under diabetic situations. For example, in patients with diabetic foot, neutrophil extracellular traps suppressed angiogenesis and delayed diabetic wound healing by inducing endothelial-to-mesenchymal transition via activating the YAP pathway [12]. YAP activation in diabetic nephropathy was associated with renal fibrosis [13]. A recent study showed that high glucose promoted vascular inflammation by activating YAP in diabetic mice [14]. However, this study only examined the effect of shear forces on glucose-treated endothelial cells without exploring upstream regulators of YAP. Meanwhile, the activation of YAP and TAZ was observed in white adipocytes from obese humans and mice [15], and YAP activation in adipose tissue induced adipocyte hypertrophy and insulin resistance [16]. However, the involvement of YAP in endothelial dysfunction under obese conditions remains unknown.

AMPK was found to act as a negative regulator of YAP activity through both the direct and indirect phosphorylation of YAP at S127 in human embryonic kidney 293 cells (HEK293) and cervical cancer HeLa cell lines [17]. AMPK phosphorylates angiomotin-like 1 (AMOTL1) and promotes LATS1 to phosphorylate YAP in HEK293A cells and HaCaT cells [18]. Xiong et al. reported that decreased phosphorylation of AMPK resulted in increased YAP expression, leading to aggravated myocardial fibrosis and inflammation in rats with diabetic cardiomyopathy [19]. However, the role of the AMPK/YAP signaling pathway in endothelial dysfunction has not been explored.

Metformin, a first-line medication for type 2 diabetes associated with obesity, lowers blood sugar levels, reduces inflammation, improves insulin sensitivity, and improves endothelial function in rodents and humans [20,21,22,23,24,25,26]. Clinical studies have shown that metformin treatment promotes weight loss, improves the metabolic profile, and increases insulin sensitivity in obese adolescents [27,28]. In vitro studies have shown that metformin increased nitric oxide production in insulin-resistant endothelial cells and suppressed vascular endoplasmic reticulum stress in diet-induced obese mice [29,30]. Metformin has recently been described to suppress YAP in multiple cancer cells through activating AMPK [31,32,33]. However, whether the AMPK/YAP axis is involved in any beneficial effects of metformin in the vascular endothelium is basically unclear.

## 2. Materials and Methods

### 2.1. Animals

The animal protocols were approved by the Animal Research Ethical Committee of the Chinese University of Hong Kong (CUHK), and the mice were handled according to the Guide for the Care and Use of Laboratory Animals published by the National Institutes of Health. Male C57BL/6 mice at 8 weeks old were kept in the CUHK Laboratory Animal Service Center with free access to food and water in a temperature-controlled environment (23 ± 1 °C) and humidity (55 ± 5%) on a 12 h light–dark cycle. Some C57BL/6 mice were fed a high-fat diet (a rodent diet with 60 kcal% fat; D12492; Research Diets, Inc., New Brunswick, NJ, USA) for 8 weeks to induce high-fat diet-induced obesity.

### 2.2. Metformin Treatment

The mice fed with a normal diet and a high-fat diet were randomly divided into 2 groups, respectively, and were administered orally with metformin (200 mg/kg/day; Sigma-Aldrich, St. Louis, MO, USA) [34] suspended in distilled water or a solvent control for 3 weeks.

### 2.3. Cell Culture and Treatment

Human umbilical vein endothelial cells (HUVECs; CC-2519; Lonza, Basel, Switzerland) were cultured in DMEM/F12 medium (11320033; Gibco™, Carlsbad, CA, USA) supplemented with 1% penicillin/streptomycin, 50 μg/mL endothelial cell growth supplement (02-102; Sigma-Aldrich, St. Louis, MO, USA), and 20% fetal bovine serum (FBS; 16000044; Gibco™, Carlsbad, CA, USA) in a humidified atmosphere with 5% CO_2_ at 37 °C. HUVECs of passage 6–8 were used. To investigate the effect of YAP on endothelial cells, the YAP plasmid was electrotransformed into HUVECs using the Basic Nucleofector Kit (VAPI-1001; Lonza, Basel, Switzerland) to overexpress YAP. Briefly, HUVECs were resuspended in a transfection mixture composed of 82 μL Nucleofector^®^ solution, 18 μL supplement solution, and 5 μg plasmid DNA (pcDNA Flag Yap1 was a gift from Yosef Shaul; Addgene plasmid # 18881; https://www.addgene.org/18881/; RRID: Addgene_18881). A specific program for HUVECs was performed for electroporation. Then, the transfection mixture was added to a 12-well plate containing fresh culture medium. After 6 h, metformin was added, and the plate was incubated for 24 h. To mimic a hyperglycemic environment in vivo, HUVECs were cultured with 30 mM glucose (HG) for 24 h in medium 199 with 10% FBS. A total of 5.5 mM glucose plus 24.5 mM mannitol (NG) was used as an osmotic pressure control. To investigate whether AMPK mediates YAP phosphorylation in endothelial cells, AMPK inhibitor compound C (10 µM; 171260; Sigma-Aldrich, St. Louis, MO, USA) was incubated together with metformin (10 mM) for 24 h in HUVECs. To test the role of JNK activation in YAP-induced inflammation, HUVECs were electrotransformed with a YAP plasmid to overexpress YAP and then treated with JNK inhibitor SP600125 (10 µM; HY-12041; MedChemExpress, Monmouth Junction, NJ, USA) for 4 h. To demonstrate that JNK is the downstream effector of YAP, HUVECs were pre-treated with metformin (10 mM) for 24 h, followed by different concentrations (0, 4, 20, 100 ng/mL) of a JNK activator, phorbol 12-myristate 13-acetate (PMA; P1585; Sigma-Aldrich, St. Louis, MO, USA), for 5 h.

### 2.4. RNA-seq and Microarray Analyses

Sequencing data are available from NCBI’s Gene Expression Omnibus (GEO). Human aortic endothelial cells (HAECs) were obtained from diabetic and non-diabetic patients (*n* = 3 per group) at autopsy (GSE77108). Fibrovascular membranes (FVMs) were surgically removed from normal humans (*n* = 3) and patients with diabetic retinopathy (*n* = 6) (GSE60436). Coronary artery segments of atherosclerotic lesions were from diabetic-hypercholesterolemic (*n* = 10 lesions) and hypercholesterolemic (*n* = 13 lesions) swine (GSE162391). For RNA-seq, Fastqc and trim galore were used to perform quality control. The sequences were mapped to the human genome (GRCh38 release 109) for the human samples and to the Sus Scrofa reference genome (V.11.1) for the pig samples using Hisat2 [35]. The tool featureCounts [36] was used to generate a matrix of read counts per gene, and DESeq2 (v1.38.1) software [37] was then used to determine differential gene expression. For the microarrays, the raw gene expression data were normalized, and differential gene expression was selected using the limma package [38]. The significance was defined as *p* < 0.05. Additionally, gene set enrichment analysis (GSEA) was performed on the gene expression profiles using the Molecular Signatures Database (MsigDB, v2023.1). |NES score| > 1 and *p* < 0.05 were used as the cutoff values for significantly enriched gene sets. The Benjamini–Hochberg procedure was used as the *p* adjustment method for the analyses above.

### 2.5. Functional Assay by Wire Myograph

The C57BL/6 mice were sacrificed by CO_2_ anesthesia, and the mouse aortas or mesenteric resistance arteries were dissected out and placed in oxygenated ice-cold Krebs solution (NaCl: 119 mM, KCl: 4.7 mM, CaCl_2_: 2.5 mM, MgCl_2_: 1 mM, NaHCO_3_: 25 mM, KH_2_PO_4_: 1.2 mM, and D-glucose: 11 mM). Ring segments of the thoracic aortas or mesenteric resistance arteries (~2 mm in length) were suspended between two tungsten wires (40 µm in diameter) in organ chambers (Multi Myograph System) filled with 5 mL oxygenated Krebs solution at 37 ℃. The changes in the isometric tone of the rings were recorded by wire myograph (Danish Myo Technology, Aarhus, Denmark). The rings were stretched to an optimal baseline tension of 3 mN (thoracic aortas) and 2 mN (mesenteric resistance arteries) and kept for equilibration for 1 h. The rings were contracted with a 60 mM KCl solution and rinsed in Krebs solution. Endothelium-dependent relaxations induced by acetylcholine (Ach, 10 nM to 10 µM, Sigma-Aldrich, St. Louis, MO, USA) or insulin (0.1 µM to 5 µM, Sigma-Aldrich, St. Louis, MO, USA) were determined in phenylephrine (Phe, 3 µM; Sigma-Aldrich, St. Louis, MO, USA) -precontracted rings. The relaxation was presented as a percentage reduction of phenylephrine-induced contraction [39,40].

### 2.6. Ex Vivo Culture of Mouse Arteries

For the ex vivo studies, thoracic aortas or mesenteric resistance arteries were cut into ring segments (~2 mm in length). All the segments were cultured in a 24-well plate with Dulbecco’s Modified Eagle’s Media (DMEM; 12320032; Gibco, Carlsbad, CA, USA) containing 1% penicillin/streptomycin and 10% FBS and kept at 37 °C and 5% CO_2_ in an incubator. To explore whether metformin ameliorates YAP activation-induced endothelial dysfunction, 1 μL of YAP adenovirus (10^10^ plaque-forming units) and GFP control adenovirus were added to the corresponding wells. After 12 h of incubation, metformin was added and the wells were incubated for another 24 h. To study whether metformin ameliorates insulin resistance, metformin and/or high glucose were added to the culture medium for 24 h. After incubation, the arterial rings were transferred to oxygenated Krebs solution, suspended in myograph for functional studies or kept for western blotting (4 aortic rings were merged into one sample).

### 2.7. ROS Determination by Dihydroethidium (DHE) Staining

Intracellular ROS generation was measured by DHE fluorescence staining. Briefly, HUVECs were washed in phosphate-buffered saline and incubated with DHE (5 μM; Invitrogen, Carlsbad, CA, USA) at 37 °C for 15 min. Hoechst 33324 was used to stain the nucleus. The DHE fluorescence was recorded by a fluorescence microscope [39].

### 2.8. Western Blotting

The mouse aorta homogenates and cells were lysed in RIPA lysis buffer (20-188; Millipore Corp., St. Louis, MO, USA) containing a Complete Protease Inhibitors cocktail (Sigma-Aldrich, St. Louis, MO, USA) and phosSTOP phosphatase inhibitor (Roche, Switzerland). The protein concentrations were measured using a BCA protein assay kit (Pierce Biotechnology, Waltham, MA, USA). An equal amount of protein was loaded onto SDS-polyacrylamide gel electrophoresis gels and transferred to polyvinylidene difluoride membranes (Millipore Corp., Burlington, VT, USA). The membranes were blocked with 3% BSA for 1 h, followed by incubation with specific primary antibodies: rabbit anti-YAP (1:1000, 14074), rabbit anti-pYAP(Ser127) (1:1000, 4911), rabbit anti-pAkt(Ser473) (1:1000, 4060), rabbit anti-Akt (1:1000, 9272), rabbit anti-pAMPKα(Thr172) (1:1000, 2535), rabbit anti-AMPKα (1:1000, 2532), rabbit anti-pSAPK/JNK(Thr183/Tyr185) (1:1000, 9251), rabbit anti-SAPK/JNK (1:1000, 9252), rabbit anti-reduced glyceraldehyde-phosphate dehydrogenase (GAPDH; 1:2000, 2118, all from Cell Signaling Technology, MA, USA), mouse anti-eNOS/NOS Type III (1:1000, 610297), mouse anti-eNOS(pS1177) (1:1000, 612392, all from BD Transduction Laboratory, San Diego, CA, USA), rabbit anti-CTGF (1:1000; ab6992), rabbit anti-CYR61 (1:1000; ab24448), and rabbit anti-VCAM1 (1:1000, ab134047, all from Abcam, Cambridge, UK). After overnight incubation, the membranes were incubated with horseradish peroxidase-conjugated anti-rabbit or anti-mouse secondary antibodies for 2 h and visualized by enhanced chemiluminescence (Cell Signaling Technology, MA, USA). The target protein levels were quantified using ImageJ and normalized to the internal control.

### 2.9. Quantitative Real-Time PCR Analysis

The total RNA was extracted from the mouse aorta homogenates and cells using TRIzol Reagent (9109; Takara, Japan). cDNA was synthesized using PrimeScript™RT Master Mix (0036A; Takara, Japan). Quantitative PCR was performed with equal amounts of cDNA in the ABI Vii7 system (Applied Biosystems, Waltham, MA, USA). The relative gene expression levels were determined using the 2^−ΔΔCt^ method, and GAPDH was used as an endogenous control. The primers used for reverse-transcriptase PCR are listed in Table 1.

### 2.10. Data Analysis and Statistics

The results were analyzed using GraphPad Prism 8 software (8.0.1) and are presented as the means ± SD. The statistical significance was determined using the unpaired t-test or non-parametric Mann–Whitney test for two groups comparison. Multiple comparisons were analyzed by one-way analysis of variance, followed by the Bonferroni multiple comparison test. A value of *p* < 0.05 was taken as statistically significant.

## 3. Results

### 3.1. High Glucose-Induced YAP Activation Accelerates Vascular Injury

To study the role of YAP signaling in high glucose-induced vascular injury, we analyzed the RNA-seq and microarray data of diabetic human aortic endothelial cells (HAECs) and fibrovascular membranes (FVMs) from the GEO database. We found significant activation of the YAP pathway and increased expression of YAP and its target genes, CTGF and CYR61, in the HAECs of diabetic patients compared with healthy subjects (Figure 1A,B). Similarly, in the FVMs of patients with diabetic retinopathy, the expression of the YAP target genes, CTGF and CYR61, also increased (Figure 1C,D). In addition, we analyzed coronary artery data from diabetic-hypercholesterolemic pigs. The expression of CYR61 and inflammatory factors was increased in the coronary artery lesions of diabetic pigs compared with non-diabetic pigs (Appendix A), while the AMPK pathway was significantly inhibited (Appendix A). These results show that YAP signaling is activated under hyperglycemic or diabetic conditions.

### 3.2. Metformin Improves Endothelial Function through Inhibiting YAP Signaling

To investigate if metformin protects vascular function through YAP inhibition, we first examined the effect of metformin on YAP phosphorylation and showed that YAP S127 phosphorylation was increased by metformin in a time-dependent fashion (Figure 2A,B). The expression of the YAP target protein CTGF was suppressed by 24 h treatment of metformin (Figure 2C). To verify whether metformin ameliorates YAP activation-induced endothelial dysfunction, we infected isolated mouse aortas with adenovirus for 12h to overexpress YAP and then treated them with metformin for 24 h. The results showed that the expression of YAP in aortic rings infected by YAP adenovirus significantly increased, as well as the YAP target genes CTGF and ANKRD1 (Appendix A). YAP overexpression impaired acetylcholine-induced endothelium-dependent relaxations (EDRs), which can be rescued by metformin treatment (Figure 2D,E). Correspondingly, YAP overexpression increased the expression of the YAP target genes CTGF, ANKRD1, and CYR61, which were reversed by metformin (Figure 2F). Taken together, these results show that metformin protects endothelial function against YAP overexpression in mouse aortas.

### 3.3. Metformin Reverses High Glucose-Induced YAP Activation

To investigate whether metformin protects endothelial cells under hyperglycemic conditions through inhibiting YAP signaling, we examined the effect of 30 mM glucose on YAP phosphorylation in HUVECs and found that high glucose suppressed YAP phosphorylation and increased the expression of the YAP target protein CYR61. These changes were reversed by metformin (Figure 3A–C). The phosphorylation of endothelial nitric oxide synthase (eNOS) on Ser 1177 augments endothelial function [41]. We next determined eNOS phosphorylation in HUVECs treated with high glucose and metformin. The results showed that reduced eNOS phosphorylation at Ser 1177 in high glucose-treated HUVECs was reversed by metformin (Figure 3A,D). The increased production of endothelial reactive oxygen species (ROS) participates in endothelial dysfunction associated with hyperglycemia [42]. We found that high glucose-increased endothelial ROS production was suppressed by metformin (Figure 3E,F), as also reported in [43].

To further evaluate the vasoprotective effect of metformin, mouse aortas were incubated with high glucose and metformin. As insulin resistance plays a critical role in the pathogenesis of obesity and diabetes-related vascular diseases, we also explored the effect of metformin on insulin sensitivity in mouse aortic endothelial cells. Consistent with the results in the HUVECs, high glucose inhibited YAP phosphorylation, which was reversed by metformin in mouse aortas (Figure 4A,B). Since Akt is the upstream regulator for eNOS phosphorylation [44], we measured the effect of metformin on Akt phosphorylation and found that metformin treatment restored high glucose-induced reduction in basal and insulin-induced Akt phosphorylation (Figure 4A,C). High glucose exposure decreased both basal and insulin-induced eNOS phosphorylation in mouse aortas, which was reversed by co-treatment with metformin (Figure 4A,D). To further show that metformin ameliorates insulin resistance, we found that the high glucose-induced impairment of insulin-induced relaxations in mouse mesenteric arteries was reversed by metformin (Figure 4E,F).

Taken together, these results demonstrate that metformin is effective at inhibiting high glucose-induced YAP activation and the associated endothelial dysfunction.

### 3.4. Metformin Improves Endothelial Function through AMPK-Mediated YAP Inhibition

Our previous study reported that the beneficial effects of metformin in the aortas of diet-induced obese mice can be blunted by an AMPK inhibitor compound C [30]. We found a marked increase in the level of AMPK phosphorylation in HUVECs after 12 h and 24 h of treatment with metformin (Figure 5A,B). Metformin-induced AMPK phosphorylation is in consistence with metformin-increased YAP phosphorylation, suggesting that metformin increases YAP phosphorylation, probably through AMPK activation. In addition, metformin reversed the high glucose-induced inhibition of AMPK phosphorylation in isolated mouse aortas (Figure 5C,D) and HUVECs (Appendix A). Further support comes from the observation that compound C reversed metformin-induced AMPK phosphorylation in high glucose-treated HUVECs (Figure 5E,F) and also suppressed metformin-induced YAP phosphorylation (Figure 5E,G), thus indicating that metformin improves endothelial function through promoting AMPK-mediated YAP phosphorylation. Furthermore, compound C inhibited the metformin-increased phosphorylation of Akt and eNOS in high glucose-exposed HUVECs (Figure 5E,H,I). These results show that metformin ameliorates endothelial dysfunction through AMPK-mediated YAP inhibition.

### 3.5. Metformin Reduces Vascular Inflammation through Inhibiting the YAP-JNK Pathway

To confirm whether metformin protects endothelial function also through inhibiting the YAP-JNK-inflammation cascade, we treated HUVECs with metformin and determined the mRNA levels of pro-inflammatory genes by real-time PCR. The results showed that metformin suppressed the expression of pro-inflammatory genes, E-selectin, VCAM1, CCL2, and TNFα (Figure 6A). YAP overexpression increased the expression of E-selectin, ICAM1, VCAM1, CCL2, and TNFα, which was reversed by metformin treatment in HUVECs (Figure 6B). JNK is the main activated MAPK in response to inflammatory stress, which is associated with glucose intolerance in obese mice [45]. Metformin inhibited high glucose-induced JNK phosphorylation (Appendix A; Figure 6C,D). To test the role of JNK activation in YAP-induced inflammation, HUVECs were first infected with a plasmid overexpressing YAP and then treated with a JNK inhibitor, SP600125, for 4 h. The results showed that the JNK inhibitor reversed the YAP overexpression-induced expression of E-selectin and ICAM1 (Figure 6E). To substantiate that JNK is the downstream effector of YAP, we treated HUVECs with different concentrations of a JNK stimulator, phorbol 12-myristate 13-acetate (PMA), in the absence or presence of metformin and found that metformin failed to suppress PMA-induced expression of inflammation in HUVECs (Appendix A), thus suggesting that metformin reduces vascular inflammation through the inhibition of YAP without a direct effect on the JNK activity.

### 3.6. Metformin Reverses HFD-Induced YAP Activation and Vascular Inflammation

C57BL/6 mice were fed a high-fat diet for eight weeks to induce obesity. The mice were treated with metformin (200 mg/kg/day) or a vehicle for three weeks. After that, the mouse aortas were isolated for western blotting and real-time PCR. Metformin treatment reversed the HFD-induced reduction of AMPK phosphorylation (Figure 7A,B). Importantly, YAP signaling was activated by HFD feeding, which was reversed by metformin administration (Figure 7A,C). In addition, metformin suppressed the mRNA expression of the YAP target genes CTGF, ANKRD1, and CYR61 (Figure 7D). Metformin treatment also reversed the HFD-induced reduction of eNOS phosphorylation (Figure 7A,E). Furthermore, HFD increased JNK phosphorylation and VCAM1 protein expression, which was reversed by metformin (Figure 7A,F,G). In addition, the HFD-induced mRNA expression of VCAM1 and TNFα was also inhibited by metformin (Figure 7H,I).

## 4. Discussion

Our previous report showed that the effector of the Hippo pathway, YAP, is a mechanosensory gene responsible for atherosclerotic plaque formation upon its activation [10]. However, clinical endothelial dysfunction, or atherosclerosis, rarely occurs alone and is often secondary to metabolic disorders such as obesity and diabetes. The key signaling pathways underlying hyperglycemic inflammation have not been fully elucidated.

YAP signaling is vital for vascular dysfunction. A recent study has shown that high glucose accelerates atherosclerosis by SNO-GNAI2 coupling with CXCR5, activating the HIPPO/YAP pathway [46]. YAP activation is also related to pathological mechanisms in non-vascular disorders, such as diabetic nephropathy [47,48], tumors [49], and renal fibrosis [50]. YAP signaling in Hippo-dependent and -independent pathways in non-vascular models indicates that unknown mechanisms of YAP regulation may exist in hyperglycemia-associated vascular injury. Further research on the vascular YAP pathway could provide a better understanding of the role of hyperglycemia or other metabolic disorders in vascular injury. Based on our previous study, metformin treatment reversed the obesity-associated impairment of endothelium-dependent relaxations [30], and we further explored new mechanisms underlying metformin-induced vascular benefits.

In this study, through mining and analyzing human vascular transcriptome data, we found the high glucose-induced activation of YAP in vascular endothelial cells and the increased expression of YAP target genes. The expression of the YAP target gene CTGF was significantly elevated, and the AMPK pathway was inhibited in diabetic pig aortic tissues, which was consistent with our findings in both the cell and animal experiments. Next, we examined whether metformin, as an activator of AMPK, can reverse the pathological process of high glucose-treated endothelial cells. Our results show that metformin-induced the phosphorylation of both AMPK and YAP in human endothelial cells, suggesting that YAP inhibition may be responsible for metformin-induced vascular beneficial effects. To reveal the role of YAP in endothelial function, we performed a functional assay on a wire myograph and showed that YAP overexpression impairs endothelial function reflected by diminished acetylcholine-induced endothelium-dependent relaxations. Metformin treatment ameliorated YAP overexpression-induced endothelial dysfunction and reversed the YAP overexpression-induced expression of YAP target genes. Metformin treatment decreased the high glucose-induced inhibition of AMPK phosphorylation, thus reversing high glucose-induced YAP activation. Furthermore, this effect of metformin was abolished by the AMPK inhibitor compound C. These new findings suggest that metformin inhibits YAP activity, likely through an AMPK-dependent mechanism, to protect endothelial function. The YAP–JNK cascade contributes to the pro-inflammatory action of YAP activation. We treated HUVECs with high glucose and found that the increased JNK phosphorylation was reversed by metformin. Meanwhile, metformin did not affect the expression of inflammatory genes induced by a JNK inducer PMA, thus suggesting that JNK is likely a downstream effector of YAP-induced inflammation. These results imply that metformin protects endothelial function by inducing YAP phosphorylation and subsequently inhibiting JNK-mediated inflammation via AMPK activation (Figure 8).

Although this study reveals a relatively complete pathway leading to vascular inflammation associated with high glucose-induced changes along the AMPK–YAP–JNK cascade, we do not rule out that metformin may also phosphorylate other sites of YAP in addition to S127, which deserves future investigation. Actually, except for the phosphorylating YAP127 site, AMPK can directly bind to and phosphorylate YAP at S61, S94, and T119 [51,52], where the S94 residue is critical for YAP–TEAD interaction in HEK293 cells [17].

## 5. Conclusions

In summary, the present study demonstrates that metformin improves hyperglycemia- and obesity-associated endothelial dysfunction by inhibiting YAP–JNK pathway-related inflammation via activating AMPK. Inhibiting endothelial YAP is effective in restoring endothelial function under obese and diabetic conditions.

## Figures and Tables

**Figure 1 antioxidants-12-01681-f001:**
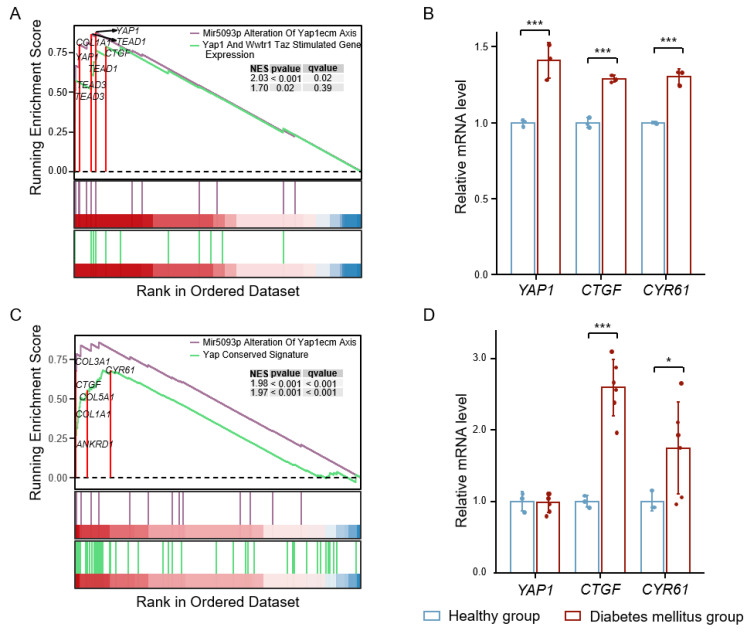
The expression of YAP signaling in diabetic HAECs and FVMs. (**A**) Normalized enrichment scores (NES) of the YAP-related pathway gene sets (|NES| > 1, *p* < 0.05) from Molecular Signatures Database (MsigDB) significantly enriched in HAECs of diabetic patients. (**B**) The relative expression levels of YAP target genes (CTGF and CYR61) in diabetic HAECs. (**C**) NES of the YAP-related pathway gene sets (|NES| > 1, *p* < 0.05) from MsigDB significantly enriched in diabetic retinopathy FVMs. (**D**) The relative expression of YAP target genes (CTGF and CYR61) in FVMs of patients with diabetic retinopathy. Data are means ± SD. * *p* < 0.05, *** *p* < 0.001.

**Figure 2 antioxidants-12-01681-f002:**
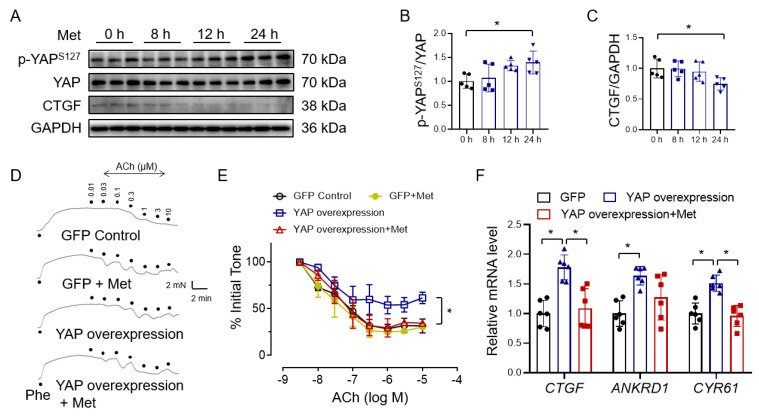
Metformin reverses YAP overexpression−induced endothelial dysfunction. (**A**) Representative immunoblots of YAP phosphorylation, total YAP, CTGF, and GAPDH in HUVECs treated with 10 mM metformin (Met) for 0, 8, 12, and 24 h. (**B**,**C**) Immunoblot quantification of YAP phosphorylation (**B**) and CTGF (**C**) in HUVECs. *n* = 5. (**D**) Representative traces. Ach: acetylcholine; Phe: phenylephrine. (**E**) ACh-induced endothelium-dependent relaxations in YAP−overexpressing aortas with or without metformin treatment. M: different concentrations of ACh (mol/L). *n* = 4. (**F**) Quantification of the mRNA expression of YAP target genes (CTGF, ANKRD1, and CYR61) in metformin−treated HUVECs after YAP overexpression. Data are means ± SD. * *p* < 0.05.

**Figure 3 antioxidants-12-01681-f003:**
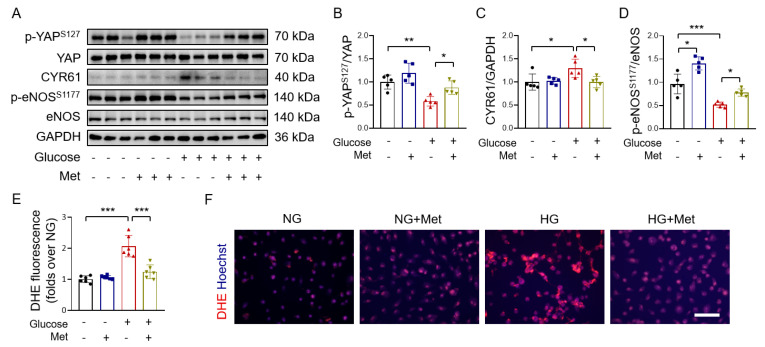
Metformin reverses high glucose-induced YAP activation in HUVECs. (**A**) Representative immunoblots of YAP phosphorylation, total YAP, CYR61, eNOS phosphorylation, total eNOS, and GAPDH in 30 mM glucose-treated HUVECs with and without co-treatment of 10 mM metformin. (**B**–**D**) Immunoblot quantification of YAP phosphorylation (**B**), CYR61 (**C**) and eNOS phosphorylation (**D**) in HUVECs. (**E**) Quantification of DHE fluorescence in HUVECs. (**F**) Representative images of DHE staining on ROS production in HUVECs. Bar = 200 μm. Met: metformin; NG: normal glucose; HG: high glucose. Data are means ± SD. * *p* < 0.05, ** *p* < 0.01, *** *p* < 0.001.

**Figure 4 antioxidants-12-01681-f004:**
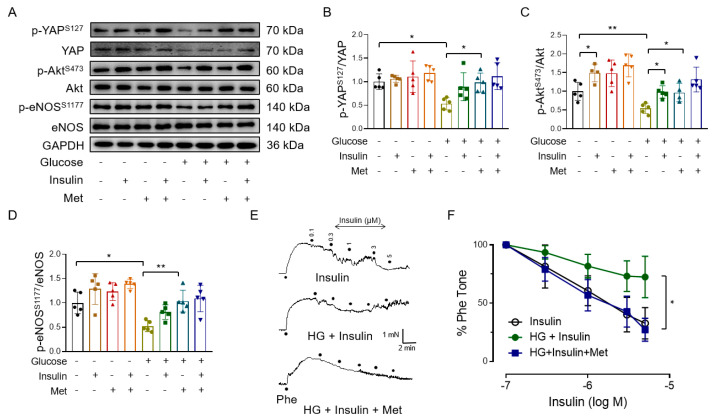
Metformin inhibits high glucose−induced YAP activation and improves insulin sensitivity in mouse aortas. (**A**–**D**) Representative immunoblots (**A**) and quantification of YAP phosphorylation (**B**), Akt phosphorylation (**C**), and eNOS phosphorylation (**D**) in isolated mouse aortas treated with 30 mM glucose (HG), 10 mM metformin (Met), and 100 nM insulin. (**E**) Representative traces. Phe: phenylephrine. (**F**) EDRs in mouse mesenteric arteries treated with high glucose, insulin, and metformin. M: different concentrations of Insulin (mol/L). *n* = 5–6. Data are means ± SD. * *p* < 0.05, ** *p* < 0.01.

**Figure 5 antioxidants-12-01681-f005:**
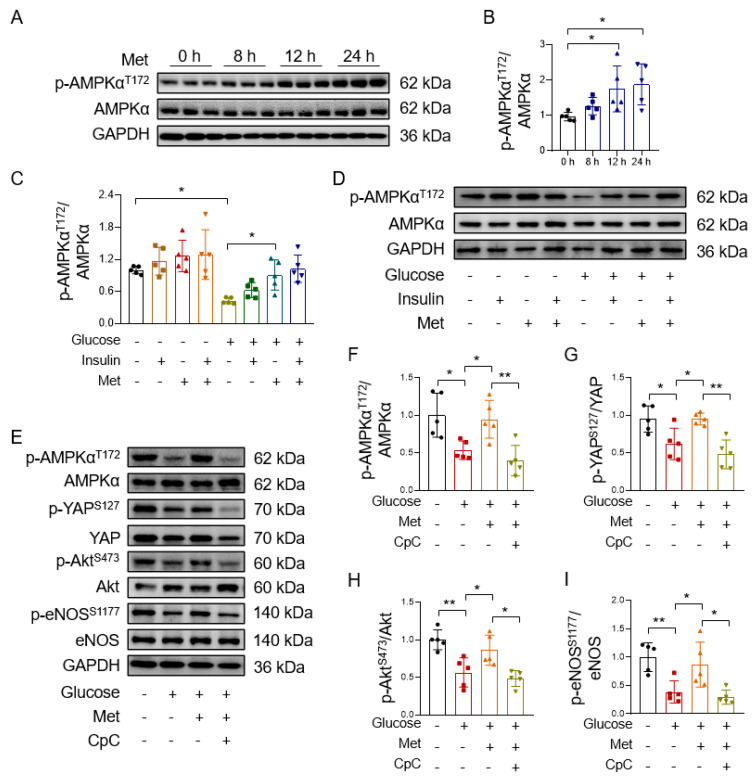
AMPK inhibition reverses the beneficial effects of metformin. (**A**,**B**) Representative immunoblots (**A**) and quantification of AMPK phosphorylation (**B**) in HUVECs treated with 10 mM metformin (Met) for 0, 8, 12, and 24 h. (**C**,**D**) Quantification of AMPK phosphorylation (**C**) and representative immunoblots (**D**) in isolated mouse aortas treated with 30 mM glucose, 10 mM metformin, and 100 nM insulin. (**E**–**I**) Representative immunoblots (**E**) and quantification of AMPK phosphorylation (**F**), YAP phosphorylation (**G**), Akt phosphorylation (**H**), and eNOS phosphorylation (**I**) in HUVECs treated with 30 mM glucose, 10 mM metformin, and 10 μM AMPK inhibitor compound C (CpC). Data are means ± SD. * *p* < 0.05, ** *p* < 0.01.

**Figure 6 antioxidants-12-01681-f006:**
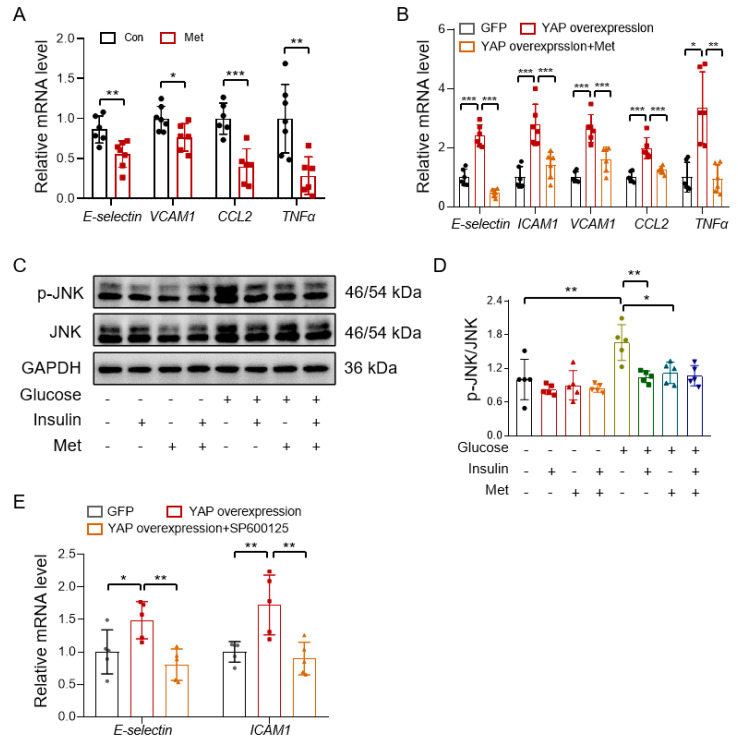
Metformin reduces vascular inflammation through inhibition of the YAP-JNK pathway. (**A**) Quantification of the mRNA expression of pro-inflammatory genes (E-selectin, VCAM1, CCL2, and TNFα) in HUVECs treated with metformin (Met) for 24 h. (**B**) Quantification of the mRNA expression of pro-inflammatory genes (E-selectin, ICAM1, VCAM1, CCL2, and TNFα) in HUVECs treated with metformin after YAP overexpression. (**C**,**D**) Representative immunoblots (**C**) and quantification of JNK phosphorylation (**D**) in isolated mouse aortas treated with 30 mM glucose, 10 mM metformin, and 100 nM insulin. (**E**) Quantification of the mRNA expression of pro-inflammatory genes E-selectin and ICAM1 in HUVECs treated with JNK inhibitor SP600125 (10 µM) after YAP overexpression. Data are means ± SD. * *p* < 0.05, ** *p* < 0.01, *** *p* < 0.001.

**Figure 7 antioxidants-12-01681-f007:**
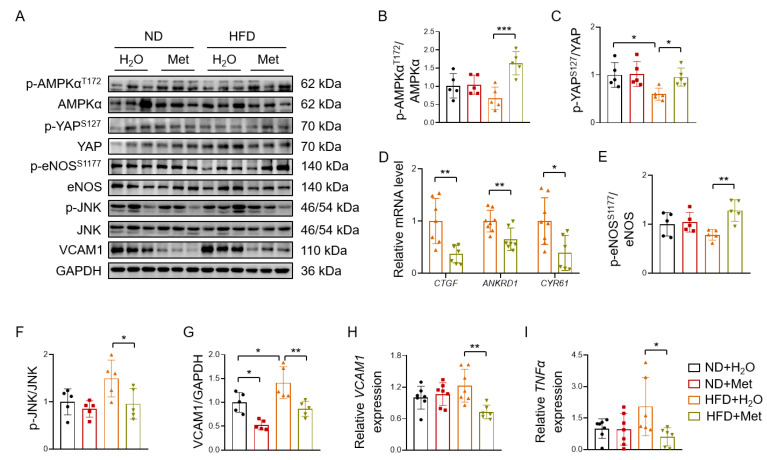
Metformin reverses HFD-induced YAP activation and vascular inflammation in mouse aortas. (**A**) Representative immunoblots of AMPK phosphorylation, total AMPK, YAP phosphorylation, total YAP, eNOS phosphorylation, total eNOS, JNK phosphorylation, total JNK, VCAM1, and GAPDH in isolated aortas from mice fed a normal diet (ND) or high-fat diet (HFD) and treated with 200 mg/kg/day metformin. (**B**,**C**) Immunoblot quantification of AMPK phosphorylation (**B**) and YAP phosphorylation (**C**). (**D**) Quantification of the mRNA expression of YAP target genes (CTGF, ANKRD1, and CYR61) in mouse aortas treated with HFD and metformin (Met). (**E**–**G**) Immunoblot quantification of eNOS phosphorylation (**E**), JNK phosphorylation (**F**), and VCAM1 (**G**) in isolated aortas from mice fed normal or high-fat diets and treated with 200 mg/kg/day metformin. (**H**,**I**) Quantification of the mRNA expression of pro-inflammatory genes, VCAM1 (**H**), and TNFα (**I**). Data are means ± SD. * *p* < 0.05, ** *p* < 0.01, *** *p* < 0.001.

**Figure 8 antioxidants-12-01681-f008:**
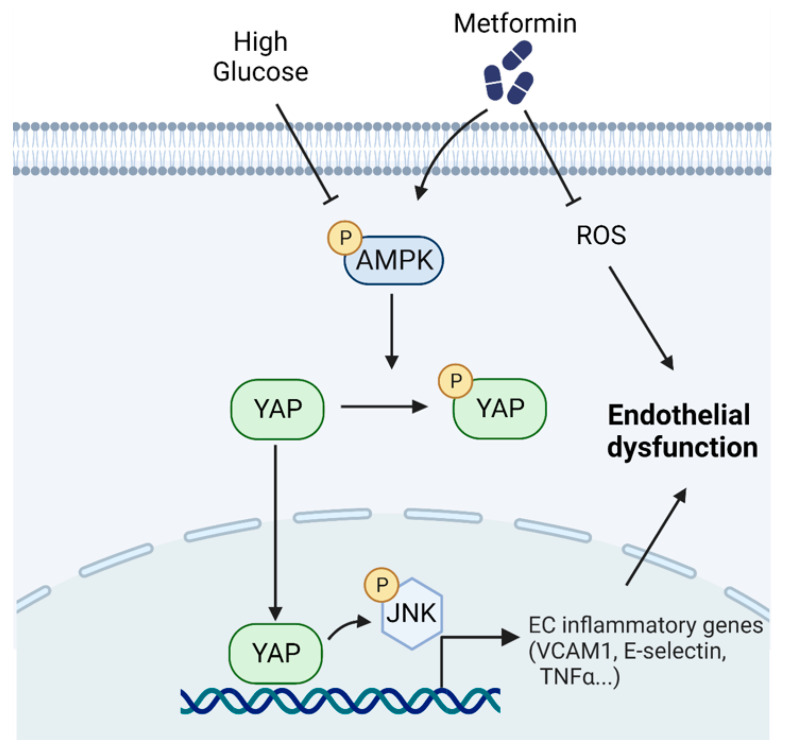
Schematic diagram of AMPK-dependent YAP inhibition mediates the protective effect of metformin against obesity-associated endothelial dysfunction.

**Table 1 antioxidants-12-01681-t001:** Primers for quantitative real-time PCR.

Primer	Sequence (5′–3′)
Human CTGF forward	ACCGACTGGAAGACACGTTTG
Human CTGF reverse	CCAGGTCAGCTTCGCAAGG
Human CYR61 forward	TGAAGCGGCTCCCTGTTTT
Human CYR61 reverse	CGGGTTTCTTTCACAAGGCG
Human ANKRD1 forward	AGAACTGTGCTGGGAAGACG
Human ANKRD1 reverse	GCCATGCCTTCAAAATGCCA
Human E-selectin forward	TGTGGGTCTGGGTAGGAACC
Human E-selectin reverse	AGCTGTGTAGCATAGGGCAAG
Human ICAM1 forward	TTGGGCATAGAGACCCCGTT
Human ICAM1 reverse	GCACATTGCTCAGTTCATACACC
Human VCAM1 forward	CAGTAAGGCAGGCTGTAAAAGA
Human VCAM1 reverse	TGGAGCTGGTAGACCCTCG
Human CCL2 forward	TCAAACTGAAGCTCGCACTC
Human CCL2 reverse	TGGGGCATTGATTGCATCTGG
Human TNFα forward	CCTCTCTCTAATCAGCCCTCTG
Human TNFα reverse	GAGGACCTGGGAGTAGATGAG
Human CXCL1 forward	CTGGCTTAGAACAAAGGGGCT
Human CXCL1 reverse	TAAAGGTAGCCCTTGTTTCCCC
Human GAPDH forward	TTCGTCATGGGTGTGAACCA
Human GAPDH reverse	TGATGGCATGGACTGTGGTC
Mouse CTGF forward	TCAACCTCAGACACTGGTTTCG
Mouse CTGF reverse	TAGAGCAGGTCTGTCTGCAAGC
Mouse CYR61 forward	GCCGTGGGCTGCATTCCTCT
Mouse CYR61 reverse	GCGGTTCGGTGCCAAAGACAGG
Mouse ANKRD1 forward	GCTGGTAACAGGCAAAAAGAAC
Mouse ANKRD1 reverse	CCTCTCGCAGTTTCTCGCT
Mouse TNFα forward	GCGGTGCCTATGTCTCAGC
Mouse TNFα reverse	CACTTGGTGGTTTGTGAGTGT
Mouse VCAM1 forward	GTTCCAGCGAGGGTCTACC
Mouse VCAM1 reverse	AACTCTTGGCAAACATTAGGTGT
Mouse GAPDH forward	AGGTCGGTGTGAACGGATTTG
Mouse GAPDH reverse	TGTAGACCATGTAGTTGAGGTCA

## Data Availability

The RNA-Seq raw data presented in the study are available from NCBI’s Gene Expression Omnibus (GSE77108, GSE60436, and GSE162391). The other data displayed in this study are within the manuscript and Appendix A.

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
