# Peer review of "AMPK-Dependent YAP Inhibition Mediates the Protective Effect of Metformin against Obesity-Associated Endothelial Dysfunction and Inflammation"

_antioxidants, 2023, doi:10.3390/antiox12091681_

Round 1
Reviewer 1 Report
Dear the Editor
Kang L et al reported a positive regulation of YAP-meditated signaling pathway through AMPK-dependent mechanism in endothelial cells. These authors examined the role of metformin, an established anti-diabetic agent, demonstrating that this attenuated endothelium-dependent relaxation through AMPK-YAP-JNK axis induced by high glucose-dependent reactive oxygen species. YAP overexpression was achieved by adenovirus-mediated technique. This is a well-designed study examining in vitro and in vivo role of metformin.
Major concerns:
1) It would be interesting to add reactive oxygen species in Fig. 8.
2) Number of data needs to be described in Figure legends (Fig. 2B).
Minor concerns:
1) Please correct typos in the text found in L24, L25, L27, L33 and others
2) Please describe either age or body weight of mice used in this study in Materials and Methods section.
3) Please provide each location of reagent manufacture.
Author Response
1) It would be interesting to add reactive oxygen species in Fig. 8.
Response: We thank this reviewer for the critical suggestion. Reactive oxygen species are critical for endothelial function. Our previous report revealed that metformin protected endothelial function by inhibiting ROS generation mediated by the metformin-PPAR-delta pathway (Cheang et al., 2014). As suggested by the reviewer, we included the reactive oxygen species in Fig. 8.
2) Number of data needs to be described in Figure legends (Fig. 2B).
Response: Thank you for raising this question. We have provided the number of data in figure legends for Figure 2B.
Minor concerns:
- Please correct typos in the text found in L24, L25, L27, L33 and others
Response: We apologize for the typos and have corrected the misspelling.
- Please describe either age or body weight of mice used in this study in Materials and Methods section.
Response: We have added the age of mice in the Materials and Methods section on page 2, line 92.
- Please provide each location of reagent manufacture.
Response: We have added the location of the reagent manufacture in the manuscript.
Reviewer 2 Report
The manuscript presented by Lijing Kang and coauthors represents an original study dedicated to the role of Hippo signaling pathway in endothelial dysfunction caused by hyperglycemia. They demonstrated that hyperglycemia upregulates YAP expression in patients’ samples, in vivo, ex vivo and in vitro. The demonstrated functional activity of YAP and negative consequences realized in inflammatory response, downregulated eNOS expression and impaired relaxation of aortic rings. These detrimental changes caused by activation of YAP signaling were mitigated by metformin (again in vivo, ex vivo and in vitro) which effect was “killed” by AMPK inhibition.
The study is technically sound and represents an interest for those who study metabolic disorders as describes previously unknow interactions between signaling pathways. However, before being published, authors should address following issues:
MAJOR:
1. Authors should provide more details on how their achieved overexpression of YAP in aortic rings. How long their incubated rings with adenovirus, what was the vector construct, what was used as control. They should also demonstrate that their overexpression model works, and they really have higher expression of YAP and higher expression of its targets (to show that functional protein is overexpressed).
2. Fig. 2 D,E. The comparison of aortic rings overexpressing YAP with aortic rings overexpressing YAP plus metformin is not informative alone. We don’t know if YAP overexpression anyhow affects aortic ring relaxation, as well as we don’t know if metformin improves this parameter without YAP overexpression. So without controls we don’t know if this effect anyhow connected with YAP signaling.
3. Fig 3, E,F. Authors should also use staining for nuclei (DAPI or Hoecst) to quantify the total number of cells per field of view, because assessing the DHE fluorescent alone could reflect changes in amount of cells.
4. Authors should provide more information on YAP overexpression in HUVECs, including but not limited to the overview of the plasmid, its concentration, time of incubation, etc.
5. What the rationale for using only male mice in experiment?
6. Fig.8 shows physical binding of YAP/TAZ complex to JNK. Could authors provide data or references justifying this?
Author Response
MAJOR:
- Authors should provide more details on how their achieved overexpression of YAP in aortic rings. How long their incubated rings with adenovirus, what was the vector construct, what was used as control. They should also demonstrate that their overexpression model works, and they really have higher expression of YAP and higher expression of its targets (to show that functional protein is overexpressed).
Response: We thank Reviewer 2 for the positive comments and valuable suggestions. The isolated aortic rings were cultured in a 24-well plate with DMEM (12320032; Gibco, Carlsbad, CA, USA) supplemented with 1% penicillin/streptomycin and 10% FBS, and kept in 37 °C, 5% CO2 incubator, 1 μL of YAP adenovirus (1010 plaque-forming units) and GFP control adenovirus were added to the corresponding wells. After 12 h of incubation, metformin was added and incubated for 24 h (please see the Materials and Methods section on page 4, lines 168-174 in the revised manuscript).
As suggested by the reviewer, we have now examined the efficiency of YAP adenovirus in isolated aortic rings. mRNA in isolated aortic rings was extracted after the vascular function. The results showed that the expression of YAP in aortic rings infected by YAP adenovirus has significantly increased, as well as YAP target genes CTGF and ANKRD1 (Supplementary Materials, Figure S2A).
- Fig. 2 D,E. The comparison of aortic rings overexpressing YAP with aortic rings overexpressing YAP plus metformin is not informative alone. We don’t know if YAP overexpression anyhow affects aortic ring relaxation, as well as we don’t know if metformin improves this parameter without YAP overexpression. So without controls we don’t know if this effect anyhow connected with YAP signaling.
Response: We agree with the reviewer and have now included an adenovirus vector control group (GFP Control) and a metformin control group (GFP + Met). The results showed that YAP overexpression impaired EDRs, which can be rescued by metformin treatment (Figure 2D&2E).
- Fig 3, E,F. Authors should also use staining for nuclei (DAPI or Hoechst) to quantify the total number of cells per field of view, because assessing the DHE fluorescent alone could reflect changes in amount of cells.
Response: We agree with the reviewer and have now provided the merged images of nuclear staining (Hoechst) and DHE staining (Figure 3E&3F).
- Authors should provide more information on YAP overexpression in HUVECs, including but not limited to the overview of the plasmid, its concentration, time of incubation, etc.
Response: We have added more detailed information in the Materials and Methods section on page 3, lines 110-117.
- What the rationale for using only male mice in experiment?
Response: Thank you for raising this question. Gender plays a crucial role in mouse models of diet-induced obesity. Generally, male mice are more vulnerable to diabetes than female mice and are consequently more frequently employed in diet-induced obesity studies [2, 3]. Male C57BL/6 mice are the most widely used animal model of diet-induced obesity [4]. Therefore, male mice were preferred for experiments in this study.
- Fig.8 shows physical binding of YAP/TAZ complex to JNK. Could authors provide data or references justifying this?
Response: Thank you for raising this important question. Our previous report revealed that YAP induces inflammation through JNK activation, the activation of the YAP-JNK pathway is important in vascular inflammation during the development of atherosclerosis [5]. Moreover, it has been reported that inhibiting YAP improves cardiac function in diabetic cardiomyopathy by hindering the recruitment of inflammatory cells and the production of pro-inflammatory cytokines through the inhibition of JNK phosphorylation [6]. In this study, we found that the JNK inhibitor reversed the YAP overexpression-induced expression of E-selectin and ICAM1 (Figure 6E) and metformin failed to suppress PMA-induced expression of inflammation in HUVECs (Supplementary Materials, Figure S4C&4D), thus suggesting that JNK is the downstream effector of YAP.
We apologize for the confusion regarding the connection relationship between YAP and JNK depicted in Figure 8. The schematic diagram has been adjusted.
[1] W.S. Cheang, X.Y. Tian, W.T. Wong, C.W. Lau, S.S. Lee, Z.Y. Chen, X. Yao, N. Wang, Y. Huang. Metformin protects endothelial function in diet-induced obese mice by inhibition of endoplasmic reticulum stress through 5' adenosine monophosphate-activated protein kinase-peroxisome proliferator-activated receptor delta pathway. Arterioscler Thromb Vasc Biol. 2014;34: 830-6.doi:10.1161/ATVBAHA.113.301938.
[2] B.-H. N. Of mice and women: the beta 3-adrenergic receptor leptin and obesity. Biochem Cell Biol. 1996;74: 615-22.doi:10.1139/o96-066.
[3] C.Y. Wang, J.K. Liao. A mouse model of diet-induced obesity and insulin resistance. Methods Mol Biol. 2012;821: 421-33.doi:10.1007/978-1-61779-430-8_27.
[4] J.A. Wali, D. Ni, H.J.W. Facey, T. Dodgson, T.J. Pulpitel, A.M. Senior, D. Raubenheimer, L. Macia, S.J. Simpson. Determining the metabolic effects of dietary fat, sugars and fat-sugar interaction using nutritional geometry in a dietary challenge study with male mice. Nat Commun. 2023;14: 4409.doi:10.1038/s41467-023-40039-w.
[5] L. Wang, J.Y. Luo, B. Li, X.Y. Tian, L.J. Chen, Y. Huang, J. Liu, D. Deng, C.W. Lau, S. Wan, et al. Integrin-YAP/TAZ-JNK cascade mediates atheroprotective effect of unidirectional shear flow. Nature. 2016;540: 579-582.doi:10.1038/nature20602.
[6] A. Zheng, Q. Chen, L. Zhang. The Hippo-YAP pathway in various cardiovascular diseases: Focusing on the inflammatory response. Front Immunol. 2022;13: 971416.doi:10.3389/fimmu.2022.971416.
Round 2
Reviewer 2 Report
Authors made an impressive work addressing all my questions.